# Differences in the Prevalence of Clinical Adjacent Segment Pathology among Continents after Anterior Cervical Fusion: Meta-Analysis of Randomized Controlled Trials

**DOI:** 10.3390/jcm10184125

**Published:** 2021-09-13

**Authors:** Young-Woo Chung, Sung-Kyu Kim, Yong-Jin Park

**Affiliations:** 1Department of Orthopedic Surgery, Gwangju Veterans Hospital, 99 Cheomdanwolbong-ro, Gwangsan-gu, Gwangju 62284, Korea; ywchungos@naver.com; 2Department of Orthopedic Surgery, Chonnam National University Medical School and Hospital, 42 Jebong-ro, Dong-gu, Gwangju 61748, Korea; ppdragon7@hanmail.net

**Keywords:** cervical vertebrae, anterior spinal fusion, adjacent segment pathology, continent, meta-analysis

## Abstract

Development of adjacent segment pathology leading to secondary operation is a matter of concern after anterior cervical discectomy and fusion (ACDF). Some studies have reported anatomic difference between races, but no epidemiological data on prevalence of clinical adjacent segment pathology (cASP) among races or continents has been published. The purpose of this study was to compare the prevalence of cASP that underwent surgery after monosegmental ACDF among continents by meta-analysis. MEDLINE, EMBASE, and Cochrane Library with manual searching in key journals, reference lists, and the National Technical Information Service were searched from inception to December 2018. Twenty studies with a total of 2009 patients were included in the meta-analysis. We extracted the publication details, sample size, and prevalence of cASP that underwent surgery. A total of 15 papers from North America, three from Europe, and two from Asia met the inclusion criteria. A total number of 2009 patients underwent monosegmental ACDF, and 113 patients (5.62%) among them had cASP that underwent surgery. The rate of cASP that underwent surgery was 4.99% in the North America, 3.65% in the Europe, 6.34% in the Asia, and there were no statistically significant differences (*p* = 0.63). The current study using the method of meta-analysis revealed that there were no significant differences in the rate of cASP that underwent surgery after ACDF among the continents.

## 1. Introduction

Anterior cervical discectomy and fusion (ACDF) has proven its effectiveness in the treatment of cervical disease in numerous long-term follow-up studies [1,2,3]. This technique involves direct decompression of the neural elements and stabilizes the affected motion segments. However, despite the benefits of ACDF, associated complications such as surgical site hematoma, damage to the neuro-vascular structures, esophageal injury, airway obstruction, and adjacent segment pathology (ASP) are frequently seen. ASP is a long-term complication after cervical fusion and a matter of grave concern until now. The pathogenesis of this complication is still a matter of debate. The etiology of ASP is most likely multifactorial and has been extensively studied in countless studies.

Previous investigators found differences in morphology of the cervical vertebral among races [4,5]. A research paper quantitatively investigated cervical morphology between Chinese and White men and reported significant differences [6]. We have wondered whether these anatomical differences affect etiology of ASP. However, to our knowledge, no epidemiological data on clinical adjacent segment pathology (cASP) leading to secondary surgery has been reported.

The original purpose of this study was to compare the prevalence of cASP among races by meta-analysis. However, we could not find any articles or research data about differences by race. Therefore, the changed purpose of this study was to compare the prevalence of cASP that underwent surgery after ACDF among continents by meta-analysis.

## 2. Materials and Methods

### 2.1. Search Strategy

Through an electronic search by two clinical investigators, we identified all randomized controlled trials (RCT) published in English up to December 2018, which involved ACDF for symptomatic cervical disc disorder. Electronic searching source included PUBMED, MEDLINE, and EMBASE. The searches used a combination of keywords describing technical procedures and anatomical features and pathology, including both MESH terms and free text words. Searches included the following keywords: “anterior cervical discectomy and fusion”, “anterior fusion”, “cervical spine”, “adjacent segment degeneration”, “adjacent segment disease”, “adjacent segment pathology”, and “randomized controlled trials”.

### 2.2. Inclusion Criteria

Eligible articles included the following required conditions: (1) studies published in English; (2) articles with randomized controlled trials; (3) ACDF performed at only 1 level for symptomatic radiculopathy; (4) minimum of 2 years postoperative follow-up duration of the included patients; and (5) articles with existence of cASP that underwent surgery after ACDF.

### 2.3. Data Extraction and Outcome Measures

A standard data extraction format was used to record the data. Our study extracted publication details (title, authors, year of publication, institution, and location of study), sample size, follow-up period, fusion material, and prevalence of cASP that underwent surgery. We considered that the continent to which the patient belongs was determined based on the location of the institution where the study was conducted. We calculated the prevalence of cASP that underwent surgery with 95% confidence intervals for each study and the overall prevalence calculated with weighted average of summary statistics through meta-analysis. The heterogeneity of data was evaluated using the I^2^ statistic. This statistic aims to assess the impact of heterogeneity on the meta-analysis [7]. The rates were defined as I^2^ < 30% as low heterogeneity, I^2^ = 30~60% as moderate heterogeneity, and I^2^ > 60% as high heterogeneity.

### 2.4. Statistical Analysis

All statistical analyses were performed using the R software (Version 3.5.1; R Foundation, Indianapolis, IN, USA). For dichotomous outcomes, odds ratios (ORs) and 95% confidence intervals (CIs) were calculated, while standardized mean differences (SMDs) and 95% CIs were derived for continuous outcomes. SMD was conducted over weighted mean difference, because different measurement indexes that adopted different tools were used in these studies. Heterogeneity among the studies was assessed by Cochrane Handbook Q test and I^2^ statistic. A *p* < 0.05 and I^2^ > 50% were considered as significant, and heterogeneity models were applied.

## 3. Results

### 3.1. Literature Search Results

After searching with the above-mentioned keywords, we identified 616 suitable articles. After exclusion of 273 duplicated articles and 291 irrelevant articles that did not meet our inclusion criteria in the abstract, we reviewed 52 abstracts. After applying further inclusion criteria, 20 studies with total 2009 patients were finally included in the meta-analysis (Figure 1). A summary for identifying relevant studies is displayed in Table 1. A total of 15 papers were from North America, 3 from Europe, and 2 from Asia [8,9,10,11,12,13,14,15,16,17,18,19,20,21,22,23,24,25,26,27]. We extracted data of these 20 studies including supplementary details.

### 3.2. Analysis of Data

#### 3.2.1. Continents

A total of 2009 patients in 20 studies underwent ACDF and 113 patients (5.62%) among these had cASP that underwent surgery. The prevalence of cASP that underwent surgery was 4.99% in the North America, 3.65% in the Europe, and 6.34% in the Asia; the prevalence values showed no significant difference (*p* = 0.63) (Figure 2). The mean follow-up duration after ACDF was 4 years in North America, 2.33 years in Europe, and 3 years in Asia. Considering similar follow-up duration for the studies, the Asia studies showed higher cASP that underwent surgery following ACDF. However, this was not statistically significant. Comparisons of the overall data were significant heterogeneity models as the I^2^ was 48% with *p* < 0.01.

#### 3.2.2. Follow-Up Duration

We also focused on the follow-up duration after ACDF and performed subgroup analysis at a different follow-up time. Follow-up duration of 11 studies was more than 4 years, and follow-up duration of 9 studies was less than 4 years. The prevalence of cASP that underwent surgery was 5.65% in the more than 4 years follow-up group and 3.69% in less than 4 years follow-up group. However, statistical analysis showed no significant difference comparing between the follow-up duration (*p* = 0.12) (Figure 3). 

#### 3.2.3. Fusion Material

The data of fusion material were collected from 18 randomized controlled trials. There were 1935 patients enrolled, 1495 in the allograft/plate group, 342 in the cage/plate group, and 98 in the tricortical iliac bone autograft/plate group. The prevalence of cASP that underwent surgery was higher in the cage/plate group (5.17%) than the other two groups, although there was no statistically significant difference (*p* = 0.74) (Figure 4).

## 4. Discussion

### 4.1. Background

The ACDF as the most common method in treating cervical disc degeneration has been carried out for many years. The ACDF is well accepted by spine surgeons, as it provides a good relief of compression and provides stability of cervical spine. However, the greatest disadvantage of ACDF is that it reduces range of motion of the cervical spine and as a result, changes the biomechanical properties of the cervical spine that may accelerate the degeneration of adjacent segments [28,29,30]. It is supposed that ASP arises when the fusion increases load and segmental motion at the adjacent levels and causes biomechanical overload with subsequent accelerated disc degeneration [31,32,33].

In a study published by the previous author, the annual incidence of cASP after anterior cervical fusion was found to be approximately 3% per year and 25.6% at 10 years [34]. Recently, Chung et al. [35] reported that 19.2% of 177 patients having cASP after ACDF with anterior plate at a minimum 10-year follow-up. A plate-to-disc distance of less than 5 mm, spondylosis, and multilevel fusion were the predisposing factors for occurrence of cASP. The etiology of cASP after ACDF is most likely multifactorial. The plate-to-disc distance, the sagittal malalignment, incorrect needle placement and soft tissue injury at the adjacent level have been proposed as etiologic factors [35,36].

### 4.2. Setting Up of Study Topic

We wondered if, considering bone size or cervical lordosis, differences between the races could affect cASP or reoperation as a risk factor. Initially, we assumed that the patient’s race between the East and the West would have differences in cASP that needed surgery. However, we could not find any articles or research data about differences by race. Additionally, Western countries such as Germany and North America include large numbers of immigrants. We could not analyze the pure differences between races. Thus, we tried to analyze the difference between the continents instead of races. The continents were classified according to the location of an article’s corresponding institution. In addition, the criteria for cASP that needed surgery were not clear and varied from article to article. So, we changed the survey with cASP that underwent surgery.

### 4.3. Analysis of Our Results

Considering the follow-up period, Asia had higher prevalence of cASP that underwent surgery after ACDF. However, this was not statistically significant. We thought that there would be a difference in cASP that underwent surgery among continents due to operating technique and facilities, or differences of representative race of the continent such as vertebral body height, surface area of end plate, bone strength, cervical lordosis, etc. Therefore, we made the following inferences: (1) If vertebral body height is higher than other races, the probability of reoperation may be less due to large plate-to-disc distance. (2) If cervical lordosis is larger than other races, the probability of reoperation may be less due to well-maintained overall cervical alignment after ACDF. (3) If bone strength is stronger than other races, the possibility of reoperation may be less due to well-maintained alignment without subsidence until fusion. However, as mentioned above, there were only few of the articles reporting cervical end plate size of different races and no study concerning the other factors (operating technique and facilities, body height, bone strength, cervical lordosis, etc.) have been reported [4,5,6,37]. We conjectured that although there is no statistical significance, the rate of cASP that underwent surgery is higher in Asia because of less cervical lordosis and small cervical vertebra. Additionally, it is expected that cASP that needed surgery is high in the long follow-up group and cage/plate group with the largest difference compared to the bone in material property. However, the analysis of these results should be further investigated in the future.

### 4.4. Limitation of Study 

This was the first systematic review on this topic. However, there are some limitations in the meta-analysis for this study: (1) Different surgeons and surgical techniques may have caused a high heterogeneity and may have led to bias. (2) Each article did not use the same classification systems or imaging diagnostic methods as the criteria for the diagnosis of cASP. (3) There were a small number of RCTs in Asia and Europe. There is a lack of power balance of the investigated groups that were compared. Therefore, the result should be accepted with caution and further research is required to reflect the broader Asian and European population. However, we think it is meaningful that it is the first paper to compare the prevalence of cASP that underwent surgery after ACDF among continents. (4) Age, sex, and race ratio of continent that could affect the outcome were not identified in our enrolled studies. (5) Due to the limitation of the study design and information in the articles included in the study, it is not clear whether differences in regional anatomic variances or in the surgery technique may influence the prevalence of cASP.

## 5. Conclusions

Our study using the method of meta-analysis showed that, although the rate of cASP that underwent surgery after ACDF is higher in Asia, there were no statistical differences among the continents. Additionally, subgroup analysis shows that fusion material and follow up period have the influence to cASP that underwent surgery, but there were no statistically significant differences. More high-quality RCTs and race differentiated anatomical studies are needed to confirm the result and to find more clinical relevance.

## Figures and Tables

**Figure 1 jcm-10-04125-f001:**
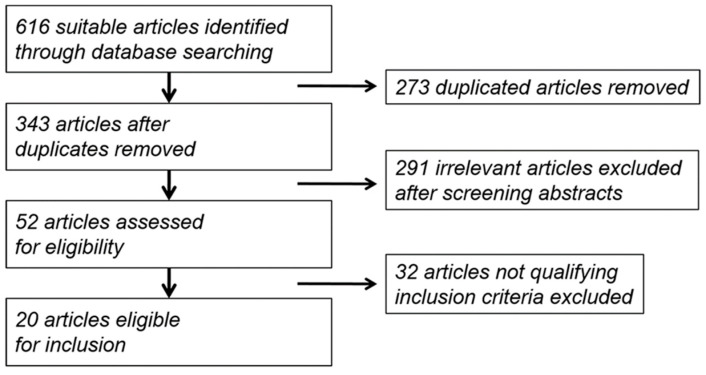
Flow diagram of the reviewed literature.

**Figure 2 jcm-10-04125-f002:**
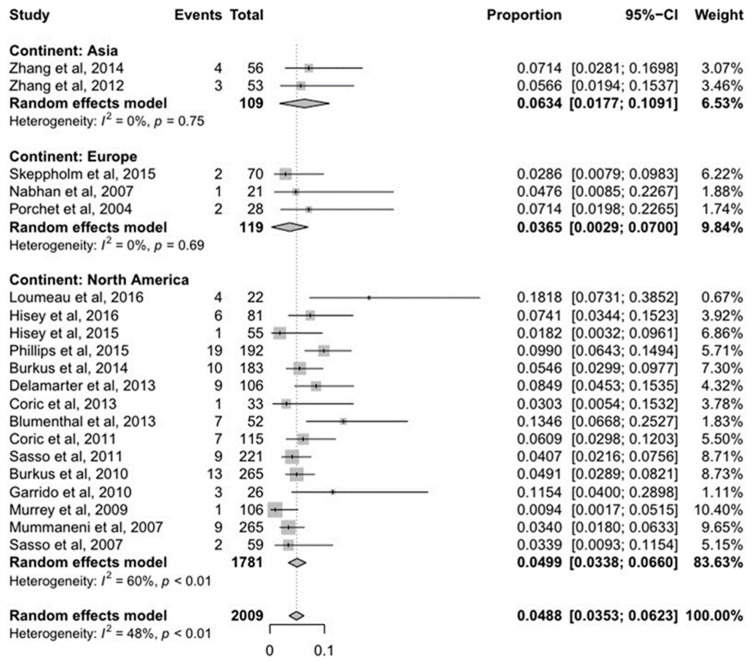
Forest plot comparing cASP that underwent surgery among the continents.

**Figure 3 jcm-10-04125-f003:**
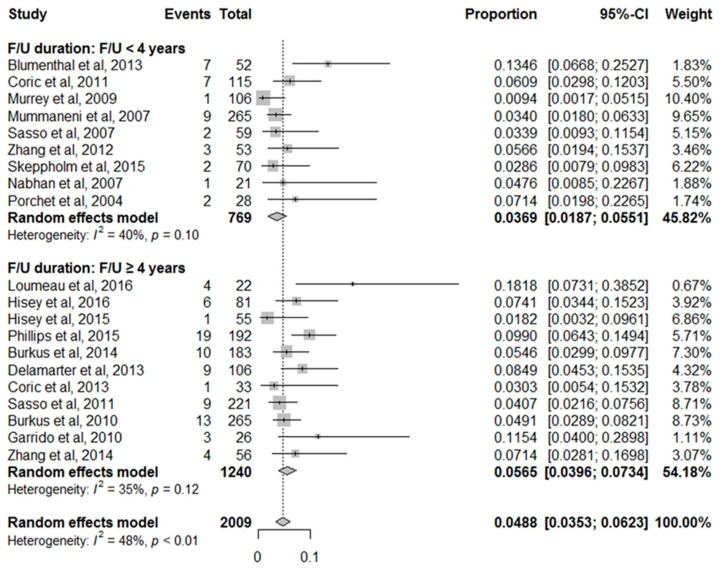
Forest plot comparing cASP that underwent surgery by follow-up duration.

**Figure 4 jcm-10-04125-f004:**
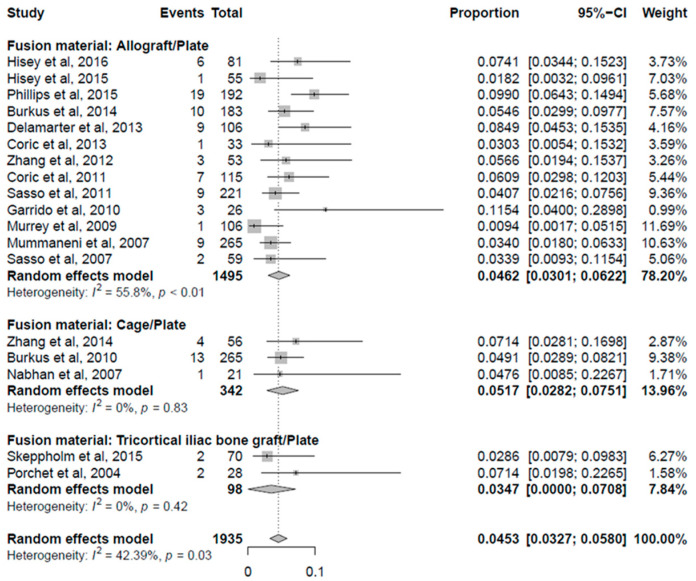
Forest plot comparing cASP that underwent surgery by fusion material.

**Table 1 jcm-10-04125-t001:** Characteristics of the studies included in the meta-analysis.

No	Authors	Journal	Year	cASP * That Underwent Surgery (%)	Country	Number of Patients	Follow-Up (Years)
1	Zhang et al. [8]	*Int. Orthop.*	2014	4 (7.1%)	China	56	4
2	Zhang et al. [9]	*Spine (Phila. Pa 1976)*	2012	3 (5.7%)	China	53	2
3	Skeppholm et al. [10]	*Spine J.*	2015	2 (2.9%)	Sweden	70	2
4	Nabhan et al. [11]	*J. Long Term Eff. Med. Implants*	2007	1 (5%)	Germany	21	3
5	Porchet et al. [12]	*Neurosurg. Focus*	2004	2 (7%)	Switzerland	28	2
6	Loumeau et al. [13]	*Eur. Spine J.*	2016	4 (18.2%)	USA	22	7
7	Hisey et al. [14]	*Int. J. Spine Surg.*	2016	6 (7.4%)	USA	81	5
8	Phillips et al. [15]	*Spine (Phila. Pa 1976)*	2015	19 (9.9%)	USA	192	5
9	Hisey et al. [16]	*J. Spinal Disord. Tech.*	2015	1 (1.8%)	USA	55	4
10	Burkus et al. [17]	*J. Neurosurg. Spine*	2014	10 (5.4%)	USA	183	7
11	Delamarter et al. [18]	*Spine (Phila. Pa 1976)*	2013	6 (5.7%)	USA	106	5
12	Coric et al. [19]	*J. Neurosurg. Spine*	2013	1 (3.0%)	USA	33	4
13	Blumenthal et al. [20]	*Spine (Phila. Pa 1976)*	2013	7 (13.5%)	USA	52	2
14	Coric et al. [21]	*J. Neurosurg. Spine*	2011	7 (6.1%)	USA	115	2
15	Sasso et al. [22]	*JBJS*	2011	9 (6.5%)	USA	138	4
16	Burkus et al. [23]	*J. Neurosurg. Spine*	2010	13 (10.2%)	USA	127	5
17	Garrido et al. [24]	*J. Spinal Disord. Tech.*	2010	3 (11.5%)	USA	26	4
18	Murrey et al. [25]	*Spine J.*	2009	1 (1.0%)	USA	106	2
19	Mummaneni et al. [26]	*J. Neurosurg. Spine*	2007	9 (3.4%)	USA	265	2
20	Sasso et al. [27]	*Spine (Phila. Pa 1976)*	2007	2 (3.4%)	USA	59	2

* cASP: clinical adjacent segment pathology.

## Data Availability

The data presented in this study are available on request from the corresponding author.

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
