# Peer review of "Differences in the Prevalence of Clinical Adjacent Segment Pathology among Continents after Anterior Cervical Fusion: Meta-Analysis of Randomized Controlled Trials"

_jcm, 2021, doi:10.3390/jcm10184125_

Round 1
Reviewer 1 Report
This manuscript is a meta-analysis of TDR outcome papers focusing on adjacent segment disease. Similar article has readily been published, however, the main objective of this one is to define the incidence by continents (Asia, EU, North A1merica). I feel very odd that spite the fact that there are only 2 studies from similar authors and country are included for Asia, and few for EU, many for North America primarily US and very old ones that have potentially high disc of the event due to the use of 1 st generation implants. Therefore, having said that there is a lack of power balance of the investigated groups that were compared, this study lacks novelty.
Author Response
Dear Editor-in-Chief & Associate Editor & Reviewer,
We thank the editor and reviewer for taking the time to review and comment on our work; the comments have greatly improved our manuscript.
The author's responses to the reviewer's comment are highlighted in yellow in the revised manuscript.
We have attempted to address the questions and comments from the reviewer to the best of our ability. We have responded to the reviewers’ comments below and have revised our manuscript.
Reviewer(s)' Comments to Author:
Reviewer #1:
This manuscript is a meta-analysis of TDR outcome papers focusing on adjacent segment disease. Similar article has readily been published, however, the main objective of this one is to define the incidence by continents (Asia, EU, North America). I feel very odd that spite the fact that there are only 2 studies from similar authors and country are included for Asia, and few for EU, many for North America primarily US and very old ones that have potentially high disc of the event due to the use of 1st generation implants. Therefore, having said that there is a lack of power balance of the investigated groups that were compared, this study lacks novelty.
- Authors’ response: What you pointed out (lack of power balance of the investigated groups that were compared) is our biggest weakness. The purpose of our study was to compare the prevalence of clinical ASP that underwent surgery after anterior cervical discectomy and fusion (ACDF) among continents by meta-analysis. Therefore, we established the inclusion criteria as follows. (1) studies published in English; (2) articles with randomized controlled trials; (3) ACDF performed at only 1 level for symptomatic radiculopathy; (4) minimum of 2 years postoperative follow-up duration of the included patients; and (5) articles with existence of cASP that underwent surgery after ACDF. There were only 20 papers that satisfied all of these criteria. A total of 15 papers were from North America, 3 from Europe, and 2 from Asia. Although there was a difference in the number of papers between continents, it would be appreciated if you consider that this is the first attempt to compare the differences between continents. Also, please consider that we compared not only ‘the differences between continents’, but also the ‘follow-up duration’ and ‘fusion material’.
Regarding your concerns, we described it in more detail as follows to '4.4. limitation of study';
“(3) There were small number of RCTs in Asia and Europe. There is a lack of power balance of the investigated groups that were compared. Therefore, the result should be accepted with caution and further research is required to reflect the broader Asian and European population. However, we think it is meaningful that it is the first paper to compare the prevalence of cASP that underwent surgery after ACDF among continents.”
Additionally, two papers in China were written by different authors.
- 8. A prospective, randomised, controlled multicentre study comparing cervical disc replacement with anterior cervical decompression and fusion.
Hao-Xuan Zhang, Yuan-Dong Shao, Yu Chen, Yong Hou, Lei Cheng, Meng Si, Lin Nie
Department of Orthopedics, Shandong University Qilu Hospital, No.107, Wen Hua Xi Road, Jinan, Shandong, 250012, People's Republic of China.
- 9. Randomized, controlled, multicenter, clinical trial comparing BRYAN cervical disc arthroplasty with anterior cervical decompression and fusion in China.
Xuesong Zhang, Xuelian Zhang, Chao Chen, Yonggang Zhang, Zheng Wang, Bin Wang, Wangjun Yan, Ming Li, Wen Yuan, Yan Wang
Division of Orthopaedics, Chinese PLA General Hospital, Beijing, China.
Once again, my coauthors and I thank you very much for your time and consideration of our submission. We hope that we have satisfactorily addressed all comments. We believe that our revisions sufficiently improved our manuscript and we hope to hear the good news from you soon that our article will be published in your journal.
Sincerely,
Corresponding author
Reviewer 2 Report
It was a pleasure to read your well written meta-analysis on the difference in the prevalence of clinical adjacent segment pathology among continents after monosegmental anterior cervical fusion. This to my knowledge is the first analysis to test for regional differences in the prevalence of this common disease. Twenty studies with total 2,009 patients were included, however there was no statistically significant difference in the prevalence of cASP depending on region, follow-up duration or fusion material. The main limitation of this study is, that it can not rule out bias and the factors leading to cASP cannot safely be identified and differentiated. Nevertheless, reporting on a rather large number of cases witch cASP, that were treated surgically from different continents adds to the novelty of the study. The below listed minor and major comments should be addressed to make this meta-analysis suitable.
Minor comments:
Introduction:
In lines 30-31 you write about “numerous long-term follow-up studies”, but only cite a single study. Please consider a review on the topic as a citation or add some more representative studies.
It can be considered an essential feature of your analysis, that monosegmental ACDF were evaluated. Please consider adding this information to the title, but at least report on it in the abstract.
Major comments:
Please state in your title, that you investigated differences in the prevalence of ASP.
Literature search result and flow diagram: Please specify “irrelevant”. What criteria were applied to decide on the relevance of the studies?
Please provide information, whether the analyzed studies measured the plate to disc distance. This is one key factor, that may lead to cASP. If so, please chose an appropriate cutoff for the plate to disc distanced to differentiate the population into two groups and analyze the prevalence of cASP based on this cutoff.
Please provide information about the criteria for the diagnosis of cASP that were used in the included study. Where all of these based on radiological findings? In plain radiography and/or MRI? Were classification systems such as the UCLA grading system or such as the classification by Hilibrand et al. used? If yes, could you differentiate between the severity of cASP? If not, please add to the limitations.
The abstract 4.3 Analysis of Our Results is highly speculative. Please support your hypotheses with literature. These hypotheses should be supported by citations:
- If vertebral body height is higher than other races, the probability of reoperation may be less due to large plate-to-disc distance.
- If cervical lordosis is larger than other races, the probability of reoperation may be less due to well maintained overall cervical alignment after ACDF.
- If bone strength is stronger than other races, the possibility of reoperation may be less due to well maintained alignment without subsidence until fusion.
In my opinion the main limitation of your study is, that you can not differentiate between reasons for cASP. The study design and apparently the information provided by the included studies does not allow to conclude, whether differences in regional anatomic variances or differences in the surgery technique may influence the prevalence of ASP. This should be added to the limitations section.
Author Response
Dear Editor-in-Chief & Associate Editor & Reviewer,
We thank the editor and reviewer for taking the time to review and comment on our work; the comments have greatly improved our manuscript.
The author's responses to the reviewer's comment are highlighted in green in the revised manuscript.
We have attempted to address the questions and comments from the reviewer to the best of our ability. We have responded to the reviewers’ comments point by point below and have revised our manuscript.
Reviewer(s)' Comments to Author:
Reviewer #2:
It was a pleasure to read your well written meta-analysis on the difference in the prevalence of clinical adjacent segment pathology among continents after monosegmental anterior cervical fusion. This to my knowledge is the first analysis to test for regional differences in the prevalence of this common disease. Twenty studies with total 2,009 patients were included, however there was no statistically significant difference in the prevalence of cASP depending on region, follow-up duration or fusion material. The main limitation of this study is, that it cannot rule out bias and the factors leading to cASP cannot safely be identified and differentiated. Nevertheless, reporting on a rather large number of cases witch cASP, that were treated surgically from different continents adds to the novelty of the study. The below listed minor and major comments should be addressed to make this meta-analysis suitable.
Minor comments:
Introduction:
- In lines 30-31 you write about “numerous long-term follow-up studies”, but only cite a single study. Please consider a review on the topic as a citation or add some more representative studies.
- Authors’ response: There were many references in our this paper, and numerous long-term follow-up studies were mostly old papers, so we selected only one representative paper. But it was our mistake.
In accordance with your advice, we have changed to the latest references.
- It can be considered an essential feature of your analysis, that monosegmental ACDF were evaluated. Please consider adding this information to the title, but at least report on it in the abstract.
- Authors’ response: In accordance with your advice, we added the “monosegmental ACDF” to the ‘Abstract’.
Major comments:
- Please state in your title, that you investigated differences in the prevalence of ASP.
Authors’ response: In accordance with your advice, we added “prevalence” to the ‘Title’.
- Literature search result and flow diagram: Please specify “irrelevant”. What criteria were applied to decide on the relevance of the studies?
- Authors’ response: We had checked the abstracts of 343 articles and excluded 291 articles that did not meet our inclusion criteria. After that, we reviewed all 52 articles to confirm that they met all of our inclusion criteria, and 20 articles were finally selected.
In accordance with your advice, we have additionally described about 'irrelevant articles' as follows;
“After exclusion of 273 duplicated articles and 291 irrelevant articles that did not meet our inclusion criteria in the abstract, we reviewed 52 abstracts.”
- Please provide information, whether the analyzed studies measured the plate to disc distance. This is one key factor, that may lead to cASP. If so, please chose an appropriate cutoff for the plate to disc distanced to differentiate the population into two groups and analyze the prevalence of cASP based on this cutoff.
- Authors’ response: As you said, 'plate to disc distance' is important for ASP occurrence. ( Ref.: Park JB, Cho YS, Riew KD. Development of adjacent level ossification in patients with an anterior cervical plate. J Bone Joint Surg Am 2005) So we also checked it in the initial stage of our Unfortunately, individual papers did not investigate this. So we couldn't study it.
- Please provide information about the criteria for the diagnosis of cASP that were used in the included study. Where all of these based on radiological findings? In plain radiography and/or MRI? Were classification systems such as the UCLA grading system or such as the classification by Hilibrand et al. used? If yes, could you differentiate between the severity of cASP? If not, please add to the limitations.
- Authors’ response: The criteria for the diagnosis of cASP were basically patients with symptoms consistent with radiological findings with ASP. However, imaging tests were different for each article, such as plain radiography and/or MRI. The classification systems were not used in all articles.
In accordance with your advice, we have additionally described to ‘4.4. Limitation of Study’ as follows;.
“(2) Each article did not use the same classification systems or imaging diagnostic methods as the criteria for the diagnosis of cASP.”
- The abstract 4.3 Analysis of Our Results is highly speculative. Please support your hypotheses with literature. These hypotheses should be supported by citations:
- If vertebral body height is higher than other races, the probability of reoperation may be less due to large plate-to-disc distance.
- If cervical lordosis is larger than other races, the probability of reoperation may be less due to well maintained overall cervical alignment after ACDF.
- If bone strength is stronger than other races, the possibility of reoperation may be less due to well maintained alignment without subsidence until fusion.
- Authors’ response: These were the ideas that led us to start this research. We wanted to investigate this, but there was no literature or data on it. We know your concern. Therefore, we have described it more clearly as below. But, if you wish to delete this sentence, we can do so.
“Therefore, we made the following inferences; (1) If vertebral body height is higher than other races, the probability of reoperation may be less due to large plate-to-disc distance. (2) If cervical lordosis is larger than other races, the probability of reoperation may be less due to well maintained overall cervical alignment after ACDF. (3) If bone strength is stronger than other races, the possibility of reopera-tion may be less due to well maintained alignment without subsidence until fusion.”
- In my opinion the main limitation of your study is, that you can not differentiate between reasons for cASP. The study design and apparently the information provided by the included studies does not allow to conclude, whether differences in regional anatomic variances or differences in the surgery technique may influence the prevalence of ASP. This should be added to the limitations section.
- Authors’ response: In accordance with your advice, we have additionally described to ‘4.4. Limitation of Study’ as follows;.
“(5) Due to the limitation of the study design and informations in the articles included in the study, it is not clear whether differences in regional anatomic variances or in the surgery technique may influence the prevalence of cASP.”
Once again, my coauthors and I thank you very much for your time and consideration of our submission. We hope that we have satisfactorily addressed all comments. We believe that our revisions sufficiently improved our manuscript and we hope to hear the good news from you soon that our article will be published in your journal.
Sincerely,
Corresponding author
Round 2
Reviewer 2 Report
Dear authors,
thank you for editing your manuscript according to the suggestions. I believe the quality of the study thus could be improved.
In my opinion the paper is suitable for publication in the present form.